# Involvement of glycogen metabolism in circadian control of UV resistance in cyanobacteria

**Koji Kawasaki**, **Hideo Iwasaki***

Department of Electrical Engineering and Bioscience, Faculty of Science and Engineering, Waseda University, Japan

* hideo-iwasaki@waseda.jp

## Abstract

Most organisms harbor circadian clocks as endogenous timing systems in order to adapt to daily environmental changes, such as exposure to ultraviolet (UV) light. It has been hypothesized that the circadian clock evolved to prevent UV-sensitive activities, such as DNA replication and cell division, during the daytime. Indeed, circadian control of UV resistance has been reported in several eukaryotic organisms, from algae to higher organisms, although the underlying mechanisms remain unknown. Here, we demonstrate that the unicellular cyanobacterium *Synechococcus elongatus* PCC 7942 exhibits a circadian rhythm in resistance to UV-C and UV-B light, which is higher during subjective dawn and lower during subjective dusk. Nullification of the clock gene cluster *kaiABC* or the DNA-photolyase *phr* abolished rhythmicity with constitutively lower resistance to UV-C light, and amino acid substitutions of KaiC altered the period lengths of the UV-C resistance rhythm. In order to elucidate the molecular mechanism underlying the circadian regulation of UV-C resistance, transposon insertion mutants that alter UV-C resistance were isolated. Mutations to the master circadian output mediator genes *sasA* and *rpaA* and the glycogen degradation enzyme gene *glgP* abolished circadian rhythms of UV-C resistance with constitutively high UV-C resistance. Combining these results with further experiments using ATP synthesis inhibitor and strains with modified metabolic pathways, we showed that UV-C resistance is weakened by directing more metabolic flux from the glycogen degradation to catabolic pathway such as oxidative pentose phosphate pathway and glycolysis. We suggest glycogen-related metabolism in the dark affects circadian control in UV sensitivity, while the light masks this effect through the photolyase function.

## Author summary

Most organisms harbor circadian clocks to adapt to daily environmental changes. It has been hypothesized that adaptation to UV radiation during the day was a driving force of the evolution of the circadian clock (known as "the flight from light" hypothesis). Thus, understanding the relationship with UV resistance is important to consider the

**Data Availability Statement:** All relevant data are within the manuscript and its Supporting Information files.

**Funding:** This study was supported in part by a Grant-in-Aid for Scientific Research (KAKENHI)

from JSPS (grant no. 18K19349, 23657138) to H.I. and a Grant-in-Aid for JSPS Fellows (grant no. 18J15016) to K.K. The funders had no role in study design, data collection and analysis, decision to publish, or preparation of the manuscript.

**Competing interests:** The authors have declared that no competing interests exist.

physiological relevance and an evolutionary origin of the circadian clock. We here demonstrate that the unicellular cyanobacterium, *Synechococcus elongatus* exhibits a circadian rhythm in resistance to UV-C light, which is higher and lower during subjective dawn and dusk, respectively. This rhythm was abolished by nullification of the clock gene cluster *kaiABC*, and the period length was changed consistently by period mutations on *kaiC*. Genetic screening revealed that nullification of clock-associating genes *sasA*, *cikA* and *rpaA*, and of a glycogen degradation enzyme gene *glgP* abolished or attenuated the UV-resistance rhythm. Combining these results with further experiments using an ATP synthesis inhibitor and strains with modified metabolic pathways, we suggest a that the circadian clock confers adaptive fitness by balancing a trade-off between glycogen-related energy metabolism and the UV-resistance property.

## Introduction

Most living organisms are exposed to dramatic environmental changes, such as day–night transitions. It is believed that the circadian clock is important for organisms to anticipate periodic environmental changes. Consistently, most organisms, ranging from cyanobacteria to higher organisms, have evolved a range of circadian oscillators that control the circadian clock.

It has been hypothesized that the circadian clock had evolved to avoid ultraviolet (UV) light-sensitive activities, such as DNA replication and cell division, during the daytime and perform such activities during the night ("the flight from light" hypothesis) [1]. The relationship between UV light and the circadian clock has been studied in various organisms from algae to higher organisms [2–4]. For example, it is thought that cryptochrome proteins, which are involved in circadian regulation in higher organisms, originate from a photolyase enzyme involved in the repair of DNA damage caused by UV irradiation [5]. Furthermore, the eukaryotic algae *Chlamydomonas reinhardtii* and *Euglena gracilis* exhibit circadian control of UV resistance, which is higher during the subjective day and lower during the subjective night [6,7]. However, in these organisms, the underlying mechanisms that drive circadian control of UV resistance remain unclear partly because the molecular mechanisms of the circadian systems in these organisms are obscure, although some components have been revealed [8].

The unicellular cyanobacterium *Synechococcus elongatus* PCC 7942 is known as the simplest model organism in circadian biology. In *Synechococcus*, three clock genes (i.e., *kaiA*, *kaiB*, and *kaiC*) form the core oscillator of the circadian system [9]. KaiC is an autokinase and autophosphatase, and its phosphorylation state oscillates with a period of 24-hour in complex with KaiA and KaiB [10–12]. A mixture of the three clock proteins KaiA, KaiB, and KaiC in the presence of adenosine triphosphate (ATP) is sufficient to reconstitute the KaiC phosphorylation cycle *in vitro* [13]. The Kai-based posttranslational clock drives genome-wide transcription rhythms under continuous light (LL) conditions [14], in which the timing information is relayed from KaiC to the SasA and RpaA two-component (His-to-Asp) output pathway [15–17]. Although many studies have reported that UV light induces physiological damage and changes to metabolic activities [18–20], no information of a direct link between the circadian clock and UV sensitivity in cyanobacteria is available. Therefore, the initial aim of the present study was to determine if circadian rhythms vary in UV resistance of *Synechococcus*.

In many organisms, UV-induced DNA damage is repaired through a photorepair reaction that is catalyzed by the photolyase enzyme, which is activated by blue light [21,22]. We found that UV resistance in *Synechococcus* fluctuates in a circadian fashion when photorepair activity was partially inhibited by dark exposure after UV irradiation (we call this experimental

condition as "UV+D", hereafter). This rhythm was dependent on the Kai-based circadian clock: disruption of the *kai* genes abolished the rhythmicity, and the period length of the UV-resistance rhythm was altered in various *kaiC* period mutant strains. This result establishes that circadian control of UV resistance is not limited to eukaryotic species but rather conserved in a wide range of species. Upon inactivation of Phr, a major DNA damage-repairing photolyase, the circadian control of UV resistance was nullified. Then, strains with altered UV-resistance rhythms were isolated by insertional transposon (Tn-5) mutagenesis. Finally, attenuation or inhibition of metabolic flux directed toward oxidative pentose phosphate pathway (OPPP) and/or glycolysis from glycogen degradation caused higher UV resistance under UV+D condition. These results suggest the presence of a possible trade-off mechanism between energy production due to glycogen metabolism and UV resistance. This interpretation would partly explain why UV resistance is not maintained at high levels but rather oscillates in a circadian fashion in many organisms.

## Results

### The circadian rhythm of UV resistance in *Synechococcus* is dependent on the *kai* gene

Initially, the time-dependent variation in UV resistance of *Synechococcus* during the LD cycle was investigated. Briefly, cells were exposed to short-wavelength UV-C light (254 nm, 500 J/ $m^2$, approximately 30 J/ $m^2$·s) every 6 h during one 12-h:12-h LD cycle after synchronization of the circadian clock by two LD cycles. Then, the cells were cultured under continuous light (LL) conditions to quantify viability against UV-C at each time point (Fig 1A, for details, see Materials and Methods). As shown in Fig 1B, growth of the WT strain was severely inhibited when cells were subjected to UV-C at dusk. When UV-C irradiation was applied at subjective midnight (ZT 18: ZT = *zeitgeber* time, referring to time in h during the LD cycle, hour 0 refers to the light onset in LD), the growth rate partially recovered to ~40% of that without UV-C (Fig 1B and 1C). The *kaiABC*-deficient (Δ*kaiABC*) strain also showed diurnal UV-C resistance, as growth was recovered more evidently when UV-C was administrated at ZT 18.

Then, we examined if the UV-C resistance rhythm persists under LL condition. Cells were exposed to UV-C for the indicated times under LL (UV+L: UV with light exposure condition) after two LD cycles and then returned to LL (Fig 1D). As shown in Fig 1E, the WT strain had no circadian rhythm of UV-C resistance, as values were almost constant throughout the circadian cycle, whereas Δ*kaiABC* cells appeared to be less UV-C resistant than the WT cells throughout LL (Fig 1E and 1F; S1A and S1B Fig). Even though the conditions before the time of UV exposure at 0, 6 and 12 hours were the same for both LD and LL conditions, the results of the growth test were different. This is because that growth was affected by the presence or absence of the subsequent darkness. Therefore, circadian rhythms were not evident under this experimental condition. Nevertheless, there are two hints of possible involvement of circadian regulation in UV resistance: (i) the magnitude of UV-C resistance differed between the WT and Δ*kaiABC* strains under the LL condition, and (ii) the growth rate after UV-C irradiation at ZT 18 under LD cycles also differed between the two strains. It should be noted that UV-C irradiation did not cause a phase shift of the circadian clock under the experimental condition (S1C Fig). Compared with the results shown in Fig 1, UV-C resistance is highly affected by darkness before and/or after UV irradiation. As shown in Fig 1B, the duration of dark exposure before or after UV irradiation differed at each time point.

To avoid this difference, UV-C resistance profiles were examined in response to LL with UV pulses immediately followed by exposure to darkness for 6 h (UV+D: UV-C with dark exposure condition, Fig 2A). Surprisingly, the results shown in Fig 2B and 2C clearly indicate

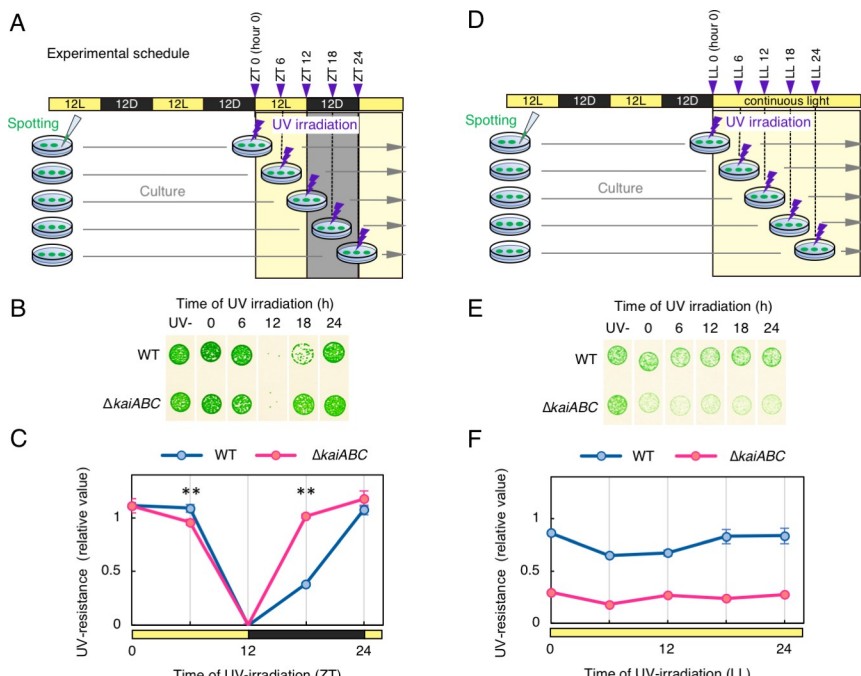

**Fig 1. Diurnal variation of UV-C resistance of *Synechococcus*.** (A) A schematic representation of the experimental schedule. Yellow and black bars indicate the light (L) and dark (D) periods, respectively. The cells were spotted onto agar plates, synchronized to two 12-h:12-h LD cycles and then irradiated with UV-C at each time point (arrowhead). (B) Growth of UV-C-irradiated cells (WT, wild-type; Δ*kaiABC*, *kaiABC*-deficient strain). Each image represents a spot assay to assess growth following UV irradiation at each time point of the 12-h LD cycle. Representative data of three independent experiments are shown. (C) Densitometric analysis of the growth test shown in Fig 1B. The timing of UV-C irradiation is shown on the horizontal axis, whereas the densitometric value of the spots as an index of cell growth and representing UV-C resistance of each strain is shown on the vertical axis. The value was normalized to that of the negative control (without UV; *n* = 3). Error bars represent standard deviation. The UV resistance in the WT and Δ*kaiABC* strains at ZT 18 significantly differ. *$P < 0.01$ (Student's t-test). (D) A schematic representation of the experimental schedule (UV+L condition). Each symbol is the same as in Fig 1A. After UV-C irradiation, the cells were exposed to continuous light. (E) Growth of the UV-C irradiated cells (WT, wild-type; Δ*kaiABC*, *kaiABC*-deficient strain). Each image represents a spot assay to assess growth following UV irradiation at each time point (upper label) under UV+L condition. Representative data of three independent experiments are shown. (F) Densitometric analysis of the growth test shown in Fig 1E. Each axis and normalization are the same as in Fig 1C. Detailed data used for figures on this article are provided in S1 Dataset.

that growth was inhibited rhythmically in the WT cells only when UV-C irradiation was administrated at subjective dusk under the UV+D condition. This result strongly suggests that *Synechococcus* exhibits circadian UV-C resistance rhythm. The "UV-"control samples shown in Fig 2A–2C were cultured under LL conditions after entraining the clock to two LD cycles. We confirmed that the time-dependent variation of growth was not caused by darkness alone, as shown in S2 Fig. When the growth of cells was compared under UV+D condition or 6-h darkness only condition without UV at different time (0, 12, and 24 h in LL), dark exposure for 6 h alone did not cause time-dependent difference in growth, whereas UV+D at hour 12 in LL significantly reduced. To avoid a possibility that this rhythmicity observed in the spot assay was due to a methodological artifact, we tested several methods for measuring cell survivability or growth. Since colony-forming units after stresses are often used for measuring survivability, we validated if rhythmicity was reproduced by counting colony-forming units in several independent experiments. As shown in S3 Fig, survivability was confirmed to fluctuate in a circadian fashion after UV-C irradiation (S3A–S3C Fig). However, considering that *Synechococcus* is a planktonic cyanobacterium, one can still question if formation of colonies might be

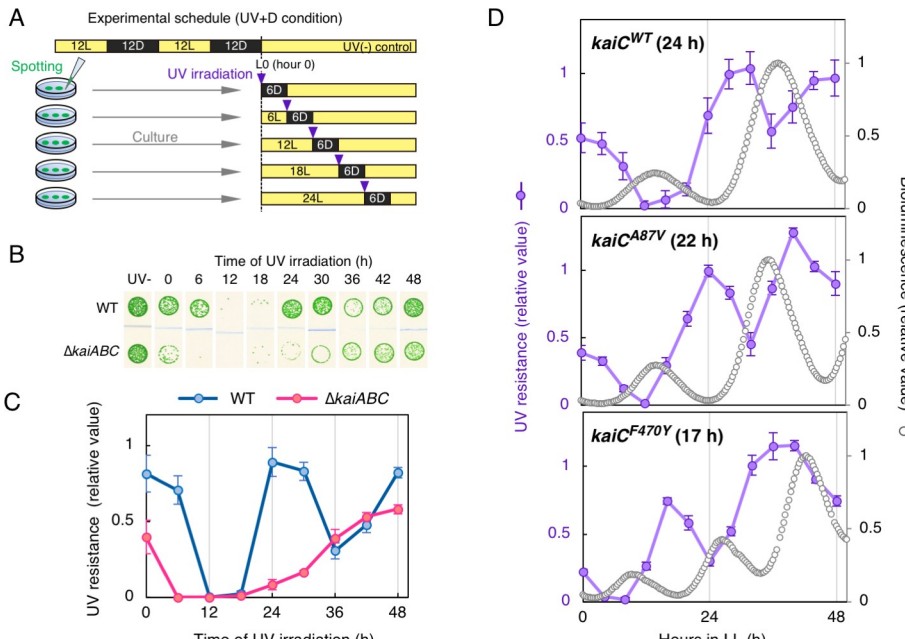

**Fig 2. *Synechococcus* shows circadian variation of resistance to UV-C immediately followed by exposure to darkness.** (A) A schematic representation of the experimental schedule (UV+D condition). Each symbol is the same as in Fig 1A. Cells were acclimated to 6 h of darkness after UV-C irradiation. (B) Growth of UV-C-irradiated cells. Each image represents a spot assay to assess growth following UV irradiation at each time point (upper label) under the UV +D condition. Representative data of three independent experiments are shown. (C) Densitometric analysis of the growth test in Fig 2B. The timing of UV irradiation is shown on the horizontal axis, and the relative UV resistance is shown on the vertical axis, as in Fig 1C (*n* = 3). Error bars represent standard deviation. (D) Time profiles of UV-C resistance in periodic mutants. Each panel represents the results in control strain (Top), *kaiC*$^{A87V}$ mutant (middle), *kaiC*$^{F470}$ mutant (bottom), respectively. Free-running period of each strain is noted in upper left of each panel. Densitometric data of UV resistance are represented with violet lines and filled circles (*n* = 3). The timing of UV irradiation is shown on the horizontal axis, and the relative UV resistance is shown on the vertical axis on the left. Bioluminescence rhythms of each strain are also shown with open circles. Each strain harbored the P$_{kaiBC}$::*luxAB* reporter cassette, and the bioluminescence rhythms were measured under continuous light (LL) condition without UV. The levels of bioluminescence are shown on the vertical axis on the right.

affected by several uncontrollable stresses and the forming units would not always correlate well with the numbers of viable cells. Therefore, some researchers prefer measuring growth in liquid media to monitor stress resistance profiles after the stress condition. Thus, we also measured growth of WT cells in liquid media after the UV+D treatment at hour 0 or 12 in the light. While cell growth was evidently confirmed after UV-C administration at hour 0, it was significantly reduced when UV was applied at hour 12, and the cell density did not recover during incubation for at least 80 h (S3D Fig). Although we tried evaluation of cell viability following UV irradiation in liquid media using vital dye, but it was difficult due to high autofluorescence of the cells. Therefore, as a second choice, we observed autofluorescence as an index of cell viability. In unicellular cyanobacteria, autofluorescence due to photosystem has been used as a convenient index of cell viability [23]. Thus, to estimate the viability of cells administrated with UV+D, we observed red autofluorescence of UV-irradiated cells under the microscope at 60 hours after the UV+D treatment. As shown in S3E Fig, despite severely limited growth after UV+D at hour L12, cells did not show defective morphology nor significant bleaching in autofluorescence, same as after UV+D at L0 (S3E Fig). Thus, UV+D at hour 12 reduced cell propagation without severe chlorophyll degradation and morphological changes. These results in liquid media are consistent with the results shown above using spotting test on solid media. Therefore, the observed rhythm in UV resistance cannot be due to methodological artifacts in

measuring the resistance. Hereafter, we mainly used the spot assays owing to its easy handling for a number of mutants and time-points at once. Under UV+D condition, we also tested the response to UV-B (312 nm) irradiation, as more natural environmental condition. Since UV-B is more abundant in the natural environment, we wanted to confirm the results obtained with UV-C are related to the behavior of cyanobacteria under natural settings. As shown in S4A and S4B Fig, the WT cells showed higher resistance against UV-B irradiated at dawn (hour 0 in the light) than that at subjective dusk (hour 12 in the light). This time dependency is essentially consistent with the experiments with UV-C irradiation. Therefore, we suggest that the results obtained with UV-C irradiation were essentially the same with UV-B.

Importantly, in the Δ*kaiABC* strain, the rhythm of UV resistance was abolished with constitutively lower levels (approximately trough levels of the WT strain, Fig 2C). Thus, UV resistance of *Synechococcus* exhibits a Kai-based circadian rhythm when UV irradiation is immediately followed by subsequent exposure to darkness. In addition, we changed timing or duration of dark exposure after UV irradiation at hour 12 in LL to test the critical parameters in the UV+D condition to affect cell growth (S4C and S4D Fig). The results showed that cell growth was permitted by light exposure for >1 h after UV irradiation before dark exposure (S4C Fig), compared with the standard UV+D condition. Moreover, the longer the exposure to darkness immediately after UV irradiation, the more severe the effect was observed on survival after UV irradiation (S4D Fig). The result shows darkness for at least 3 h was sufficient to observe the UV resistance rhythm under our experimental conditions. The efficient length of darkness can also be influenced by UV intensity.

## The period of UV resistance rhythm is consistent with the period of endogenous clock

In general, circadian rhythm refers to the phenomenon under continuous conditions. Considering UV-C irradiation and subsequent darkness as a kind of stress, the time-dependent cycle to the stress can fit the classical example of circadian rhythms. However, the above-mentioned UV-C resistance rhythm accompanies with an external dark pulse under UV+D condition. Thus, one may doubt whether this process is actually related to the circadian rhythm or is more related to difference in metabolism during light/dark cycles. In order to support that the UV-C resistance rhythm under UV+D condition is a *bona fide* free-running circadian rhythm derived from the endogenous clock, we performed experiments using previously reported short-period mutant strains mapped on *kaiC*. If the rhythm is an autonomous oscillation by the circadian clock (in other word, that is independent of the external dark pulse), the period of the UV-C resistance rhythm should fluctuate according to the lengths of endogenous (free-running) period. The *kaiC*$^{A87V}$ and *kaiC*$^{F470Y}$ mutations have been shown to shorten the free-running period of KaiC phosphorylation rhythm *in vitro* and the *kaiBC* promoter activity rhythm monitored with bioluminescence with a period length of 22 and 17, respectively [9,13]. Short period bioluminescence profiles of these mutant strains were confirmed as shown in Fig 2D. Importantly, each mutant also shortened the period of UV-C resistance rhythm and its length correlated well with their each free-running period even in the presence of the external darkness under the UV+D condition (Fig 2D). These results established that the observed UV-C resistance rhythm under UV+D condition is a *bona fide* time-dependent response derived from the endogenous clock.

## The photorepair reaction is involved in UV-C resistance of *Synechococcus*

The role of dark exposure after UV-C irradiation in the modification of UV-C resistance was investigated. Major DNA damage caused by UV irradiation results in cyclobutane pyrimidine

dimers (CPDs) formation [24,25]. CPDs are removed *via* multiple pathways, such as the photorepair and nucleotide excision repair (NER) processes [20,25]. In the unicellular cyanobacterium *Synechocystis* sp. PCC 6803, the photorepair reaction catalyzed by CPD photolyase, which is activated by visible light, plays a major role in CPD removal [26,27]. *Synechococcus* harbors the *phr* gene (*Synpcc7942_0112*), which encodes a homolog of CPD photolyase. It was reported that *Synechococcus* Phr binds to CPD-like DNA [28,29], and heterologous expression of the *Synechococcus phr* gene in *Escherichia coli* improves UV resistance [30]. Thus, the *phr* gene of *Synechococcus* is expected to encode a *bona fide* photolyase enzyme. We expected that dark exposure after UV-C irradiation (UV+D condition) would prevent the Phr-based photorepair reaction (S5A and S5B Fig for schematic conditions to be compared). In other words, we hypothesized that the results shown in Fig 2 would be interpreted as a clock-controlled response of cells against inhibition of the photorepair reaction.

When the *phr* gene was nullified, UV-C resistance of the mutant (Δ*phr*) strain was severely lowered at all time points under UV+L conditions (S5C Fig). Thus, the growth of *Synechococcus* after UV-C irradiation is greatly dependent on the photorepair pathway by CPD photolyase, as with other cyanobacteria. As expected, the Δ*phr* strain did not recover after UV-C exposure under the UV+D condition (Fig 3A).

Next, the role of Kai-based time-dependent Phr activity on the circadian variation of UV-C resistance of *Synechococcus* was investigated. According to the results of our previous DNA microarray study (14), the accumulation of *phr* mRNA level did not show a circadian rhythm. Hence, a strain was constructed in which *phr* was expressed by an isopropyl β-d-1-thiogalacto-pyranoside (IPTG)-inducible heterologous *trc* promoter with a *phr*-deficient background (Δ*phr; ox-phr* strain). The results showed that the circadian rhythm of UV-C resistance was rescued by continuous (leaky) *phr* expression even in the absence of IPTG (Fig 3A). The same result was confirmed in the presence of 100 μM IPTG (S5D and S5E Fig). These results negate the possibility that the circadian rhythm of *phr* expression is involved in the rhythmic control of UV resistance. However, it is possible that the UV-C resistance rhythm is generated by post-transcriptional regulation or posttranslational modification of Phr.

Thus, variations in DNA repair activity of cells exposed to UV-C at subjective dawn vs. dusk were examined. WT cells were exposed to UV at hour 0 or 12 in LL and then divided into three groups. The first group was immediately harvested as a UV+ control sample, whereas the second group was cultured in the light for 6 h and then harvested (light repairing sample, shown as UV+6L in Fig 3B), and the third group was harvested after acclimation to darkness for 6 h (inhibition of DNA repair sample, shown as UV+6D in Fig 3B). Genomic DNA was extracted from each group, and the CPD content was quantified using an ELISA with a specific antibody against CPD. The results confirmed that UV-C irradiation increased CPD levels by four- to fivefold, as compared to the non-UV control, regardless of the timing of UV-C administration (0 or 12 h of light) (Fig 3B). When cells were maintained in the light after UV-C exposure, the CPD level decreased to the control level, while dark exposure failed to reduce it, regardless of the timing of UV-C irradiation (Fig 3B). These observations are consistent with the results of the present study that UV resistance of *Synechococcus* is highly dependent on the Phr-mediated CPD photorepair pathway. Nevertheless, time-independent photorepair activity, as shown in Fig 3B, cannot explain the time-dependent resistance against UV (Fig 2). Thus, CPD photolyase activity following dark acclimation is not likely a major target of circadian control that gives rise to the rhythm of UV-C resistance. Instead, rhythmic control of UV-C resistance is overt when CPD photorepair is partially inhibited by the dark but is masked when CPD photorepair activity is intact without darkness.

Then, we expected that the circadian clock would control (i) UV-induced DNA damage, with the exception of the CPD and CPD photolyase-independent DNA repair processes, and/

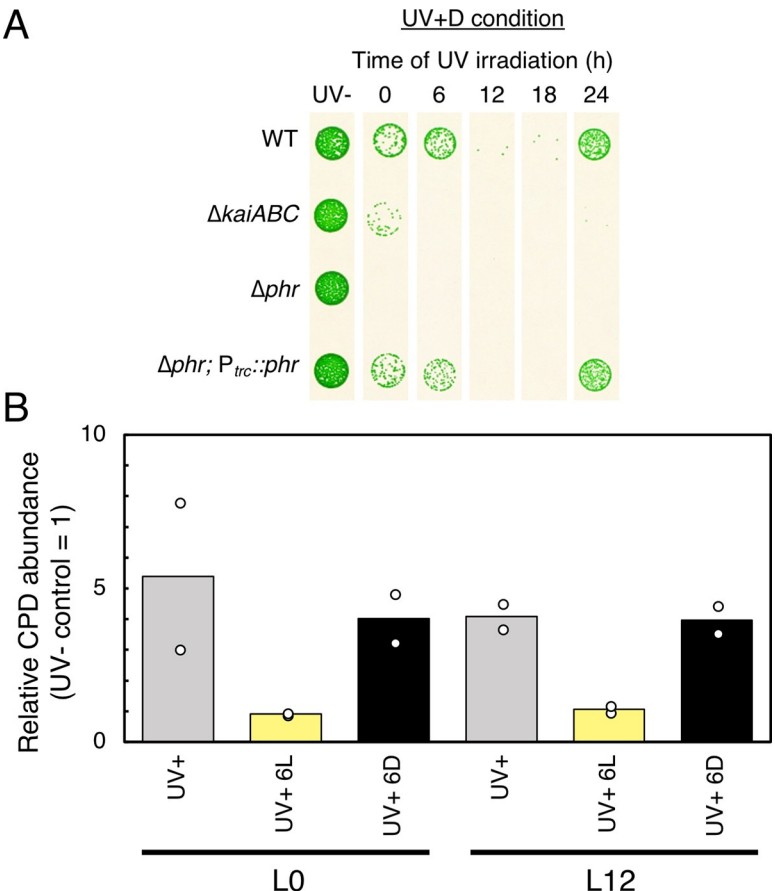

**Fig 3. Importance of DNA photorepair activity and its circadian independency.** (A) Growth of UV-C-irradiated cells of each mutant strain under the UV+D condition. Δ*phr*, *phr*-deficient strain. In the Δ*phr*; P$_{trc}$::*phr* strain, *phr* was expressed under the *trc* promoter with a *phr*-deficient background. In the Δ*phr*; P$_{trc}$::*phr* strain, ectopic *phr* expression at the leaky level without IPTG was sufficient to recover the circadian rhythm of UV-C resistance. Representative data of three independent experiments are shown. (B) Genomic DNA damage, as quantified by ELISA using a CPD-specific antibody. Genomic DNA was extracted from cells after UV irradiation at hour 0 or 12 in the light. The cells were harvested immediately after UV irradiation, (control sample: UV+) and exposed to the light for 6 h (light repairing sample: UV+6L) or the dark for 6 h (inhibiting repair sample: UV+6D). The vertical axis shows signal value obtained by CPD ELISA. In each case, the CPD level was normalized to samples without UV irradiation, which were assigned signal values of 1 (*n* = 2). Each plot indicates the result of two independent experiments, and bars indicate the mean values.

or (ii) UV-induced damage other than DNA damage and the related repair mechanism. In the case of (i), (6–4) photolyase is not likely involved, since in the *Synechococcus* genome, only the *phr* gene harbors a photolyase domain. Generally, photolyases have high substrate specificity and bind to either CPD or the (6–4) photoproduct in a structure-specific manner [31]. The crystal structure of the *Synechococcus* Phr protein has been shown to bind a CPD analog [28]. Also, the histidine residue, which is essential for the catalytic activity of (6–4) photolyase [32], is not conserved in the *Synechococcus* Phr protein.

Nucleotide excision repair (NER) is a representative photorepair-independent DNA repair mechanism. Inactivation of the *uvrA* gene, which participates in the NER process, moderately reduced UV-C resistance to ~50% of that in the wild type strain under the UV+L condition (S5F and S5H Fig). More significantly, UV-C resistance was dramatically lowered in the *uvrA* strain under the UV+D condition (S5G and S5I Fig). Nevertheless, an attenuated time-dependent variation in UV-C resistance still remained (S5I Fig, see the magnified line), suggesting

that *uvrA* is important for the magnitude of UV resistance but not essential to drive the rhythmicity. Our previous DNA microarray analysis indicated that the *uvrA* mRNA expressed in a *kai*-dependent circadian fashion as many other genes (S5J Fig, [14]). However, its amplitude is low with peak-to-trough ratio of ~1.4, which would be insufficient to drive a high amplitude protein abundance rhythm. Although we cannot rule out the possibility that clock-controlled NER somewhat supports and amplify UV resistance rhythms when CPD photolyase activity is partially inhibited by dark exposure, further studies are needed to validate this possibility.

As an alternative possibility, with the exception of the DNA repair process (case ii), the circadian clock might control the resistance mechanism against reactive oxygen species induced by UV light. We examined if sensitivity to $H_2O_2$ varied at hour 0 and 12 in the light. Cell suspensions were treated with $H_2O_2$ at each time point, incubated for 6 h either in the light or dark, and then inoculated onto solid media to quantify the growth rate. As shown in S6 Fig, the growth profiles of the cells did not significantly vary, regardless of the timing of $H_2O_2$ administration or subsequent light/dark exposure. Although it remains a room to test more reactive ROS such as superoxide or singlet oxygen, it was difficult to directly identify factors that control the circadian rhythm of UV resistance. Therefore, genetic screening was performed to identify such factors.

## Tn-5-based screening identified genes involved in UV-C resistance control

To elucidate the mechanisms underlying circadian rhythms in UV-C resistance, mutants were isolated from the randomly transposon (Tn5)-inserted mutant library, which abolished the low UV-C resistance profile, when UV was irradiated at hour 12 in the light under the UV+D condition (Fig 2A, see Materials and Methods, for experimental schedule). Of 5,000 clones that were screened, four mutant strains were isolated with greater UV resistance at hour 12 than the WT strain, thereby showing attenuated circadian variation (Fig 4A). Two mutations (mutants 2 and 3) were mapped to the coding region of *sasA* and a region upstream of *rpaA* (presumably the *rpaA* promoter region), respectively (Fig 4B). More detailed information of Tn-5 insertion sites was shown in S7A Fig. The KaiC-binding histidine kinase SasA and its cognate response regulator RpaA constitute a major circadian output mediator that drives genome-wide circadian transcriptional orchestration [15–17]. Both mutants 2 and 3 abolished the bioluminescence rhythm to monitor *kaiBC* promoter activity under continuous light conditions (S7B and S7C Fig) at constitutively low levels, as reported previously for knockout strains which deleted each gene [15,16]. These results further support that the circadian rhythm of UV-C resistance is dependent on the transcriptional output of the clock *via* the SasA-RpaA system. Disruption of *sasA* and *rpaA* has been known to arrest the transcriptional rhythm at subjective dawn-like state under standard continuous light conditions [15,16,33]. By contrast, disruption of another clock-related histidine kinase gene, *cikA*, shifts the transcriptional profile to the subjective dusk-like state, while low amplitude rhythm remains [33,34]. Thus, we prepared two strains either deficient of *sasA* or *cikA* for UV-C resistance assay under the UV+D condition. As shown in Fig 4C and 4D, UV-C resistance rhythm was abolished in both *sasA* and *cikA* strains with opposite profiles. The *sasA* mutant maintained constitutively higher UV-C resistance level (comparable to the peak level of the rhythm in the wild type strain. On the contrary, the *cikA* strain showed constitutively lowered UV-C resistance throughout the circadian cycle (comparable to the trough level of the wild type rhythm. This opposite effect is well correlated with the opposite functions of SasA and CikA: SasA activates RpaA phosphorylation to enhance transcription of genes peaking at subjective dusk, while CikA dephosphorylates RpaA [34]. Roughly speaking, above-mentioned results suggest that UV-C resistance is lowered when expression of subjective dusk genes is relatively

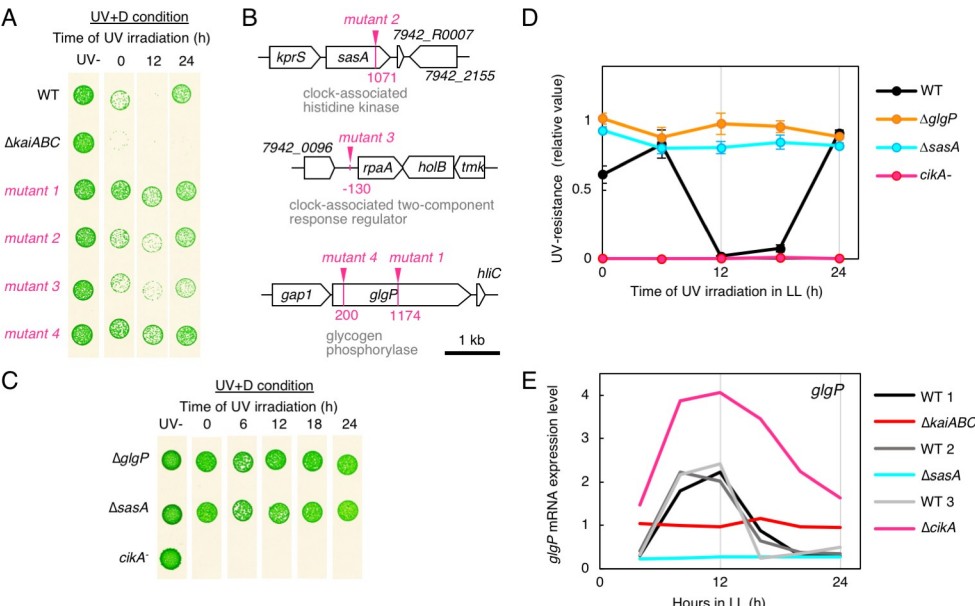

**Fig 4. Isolated mutants showed loss of circadian UV-C resistance.** (A) Effects of UV-C irradiation on isolated mutants under the UV+D condition. Each image represents a spot assay to assess growth as shown in Fig 2B. (B) Mapping of Tn-5 insertions. The Tn-5-insertion sites of each UV resistance mutant are indicated by arrowheads. The mutations were mapped onto the open reading frames (ORFs) of the *glgP* and *sasA* genes, and the upstream region of *rpaA*. The function of each gene is indicated under each gene symbol. The numbers under the arrowheads indicate the positions of insertion sites when the start of the ORF of each mutated gene is assigned a positional value of 1. (C) Growth of the UV-C-irradiated cells of each re-constructed mutant strain (Δ*glgP*, *glgP*-null strain; Δ*sasA*, *sasA*-null strain; *cikA⁻*, circadian input kinase *cikA*-deficient strain) under the UV+D condition. Each image represents the result of a spot assay to assess the growth of each mutant following UV irradiation at each time point. Representative data of three independent experiments are shown. (D) Densitometric analysis of the growth test in Fig 4C. The timing of UV irradiation is shown on the horizontal axis, and relative UV resistance is shown on the vertical axis, as in Fig 1C (*n* = 3). Error bars represent standard deviation. (E) *glgP* mRNA accumulation profile in each mutant strain. The *glgP* expression level from microarray data reported in the previous studies is shown. The microarray data were extracted for Δ*kaiABC* and the corresponding control wild type (shown here as WT1) strains from Ito *et al.* [14], for Δ*sasA* and its control (WT2) strains from data available at (NCBI, Gene Expression Omnibus databank, #GSE28430), and Δ*cikA* and its control (WT3) strains from Pattanayak *et al.* [33]. The expression data of *glgP* in each WT strain were normalized to the average of 1.

enhanced, such as any circadian time in the *cikA* and *kaiABC*-null mutant strains, and subjective dusk in the wild type strain. In both *sasA* and *rpaA* strains, expression of subjective dusk genes are constitutively suppressed and the magnitude of UV-C resistance is maintained at high levels.

The other two of four isolated mutations (mutants 1 and 4, shown in Fig 4A) which did not show attenuated UV-C resistance at hour 12 in the light were mapped to different positions of the same gene, *glgP*, which encodes glycogen phosphorylase (Fig 4B). Glycogen phosphorylase is an enzyme that degrades glycogen and produces glucose-1-phosphate [35]. To confirm the results observed with the Tn-5-insertion mutants, strain without any *glgP* coding sequence was newly constructed. This strain showed constitutively higher UV-C resistance under the UV+D conditions throughout the circadian cycle, phenocopying the *sasA* strain (Fig 4C and 4D). Interestingly, according to our previous microarray analysis [14], *glgP* is a typical high-amplitude clock-controlled gene which peaks at subjective dusk in LL, while it is constitutively maintained at moderate levels in the *kaiABC*-null mutant (Fig 4E). Accordingly, ChIP-seq analysis by Markson *et al.* [17] has identified this gene as a RpaA-binding target, which is directly activated by RpaA. Thus, *glgP* expression is strongly downregulated in either *rpaA* or

*sasA* strain, and upregulated constitutively in *cikA* strain (Fig 4E). Thus, all these results do not contradict with the hypothesis that UV-C resistance is lowered when *glgP* expression is relatively enhanced. In the wild type strain, glycogen accumulation fluctuates in a circadian fashion, with a maximum at dusk and a minimum at dawn in LL [36]. This is most likely due to anti-phasic expression and/or enzymatic activities of enzymes for glycogen degradation and synthesis. To be more precise, the expression of glycogen degradation enzyme genes such as *glgP* peaks at dusk via the SasA-RpaA-mediated pathway, while the expression of glycogen synthesis genes such as *glgC* peaks at dawn. In other words, glycogen degradation activity peaks at the time of the maximum glycogen content, while the glycogen synthetic activity is higher at the time of the minimum glycogen level. Thus, the activity of degradation and synthesis of glycogen do not coincide with the accumulation of glycogen at that time, and the fluctuation of glycogen accumulation delays by ~12 hours relative to that of the enzymatic activity. Therefore, even though glycogen content peaks at dusk, it decreases over the following 12 hours due to increased degradation activity and decreased synthetic activity.

It should be noted that nullification of either *sasA* or *rpaA* is known to reduce glycogen content, while that of *cikA* over-accumulate it *via* genome-wide transcriptional control [33,36,37]. Therefore, any of the mutations identified here would give rise to abnormalities in temporal dynamics of glycogen metabolism. Interestingly, glycogen content was lower in the *rpaA* strain, despite the fact that the maximum catalytic activity of enzymes involved in glycogen biosynthesis are not reduced the *rpaA* strain [37]. We suggest that the intracellular state of the *rpaA* strain is arrested at the dawn phase state of the wild strain, where glycogen synthetic enzyme may not catalyze its reactions since some steps for glycogen synthesis such as gluconeogenesis pathway are depleted. In the *rpaA* strain, energy charge (relative levels of ATP, ADP and AMP in the intracellular adenine nucleotide pool) strikingly drops when cells are exposed to the dark, compared with the wild type strain [37]. While restoration of energy charge by forced glucose uptake alone seems not sufficient to recover higher glycogen content [37], ATP synthesis may also be a rate-limiting factor for glycogen synthesis in the *rpaA* strain. In any cases, this background indicates that it is not easy to estimate the intracellular physiological state, including the activity of glycogen degradation and synthesis, based solely on the amount of glycogen content at each time. The link between UV resistance and glycogen metabolism suggested by the screening would be either the result of the glycogen content itself, the effect of the activity of glycogen degradation by *glgP*, or the effect of the downstream pentose phosphate circuit and the TCA cycle.

## Involvement of the glycogen metabolic gene in UV-C resistance control

We aimed to clarify the relationship between UV resistance and glycogen metabolism. A simplified diagram of the glycogen-related metabolic pathways is shown in Fig 5A. GlgP plays an important role in glycogen degradation and is involved in major metabolic pathways, such as the glycolysis and oxidative pentose phosphate pathways, which are affected by glycogen catabolism [35,38]. Thus, we evaluated whether the *glgP* strain actually lowered glycogen degradation activity under our experimental conditions. Initially, we monitored glycogen accumulation levels in the wild type strain in LL, which showed an overt circadian rhythm peaking at subjective dusk (S8A Fig) [33]. Glycogen is believed to be important as a carbohydrate reservoir, mainly for survival under ATP-limited dark condition in the absence of photosynthesis activity [36]. Therefore, we tested glycogen consumption profile during 6-h of darkness applied at subjective dawn and dusk. Dark exposure starting at hour 0 maintained glycogen content at constitutively low level (Fig 5B), comparable to the trough level of the circadian glycogen rhythm in LL (S8A Fig). On the other hand, dark exposure at hour 12

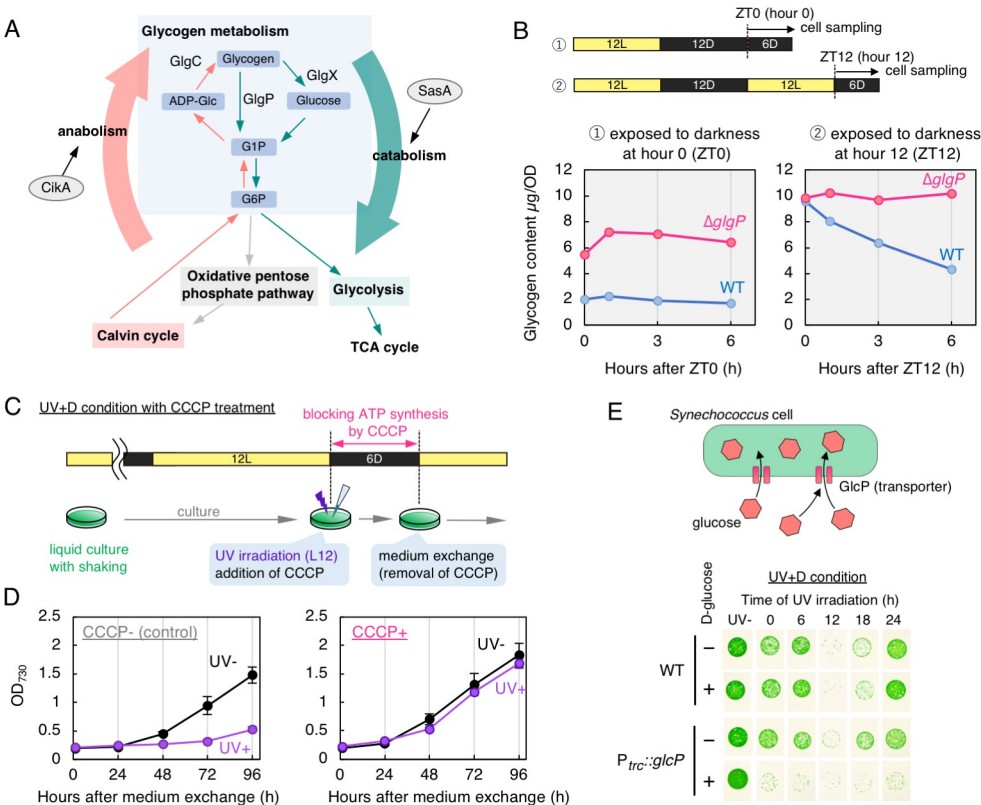

**Fig 5. Glycogen and related metabolism affect circadian control of UV-C resistance.** (A) A simplified diagram of the glycogen-related metabolic pathways (G1P, glucose-1-phosphate; G6P, glucose-6-phosphate; ADP-Glc, adenosine diphosphate glucose). GlgP is the core enzyme involved in glycogen degradation. SasA and CikA activate the catabolic and anabolic metabolic pathways, respectively, through mediating the timing information from the core oscillator to transcriptional machinery to control gene expression. (B) Quantification of glycogen content in WT and $\Delta glgP$ (*glgP*-null) strains. Cells were exposed to darkness and then harvested at each timepoint. Left and right panels show the dynamics of glycogen content when the cells were transferred into darkness at hour 0 and 12 in LL, respectively. Hours in the dark is shown on the horizontal axis, and glycogen content is shown on the vertical axis. The glycogen contents are normalized by $OD_{730}$ unit of harvested cells. (C) A schematic representation of the experiment. UV-C irradiation and dark exposure was performed at hour 12 in LL. During the darkness, CCCP was added to the culture. At the end of darkness, CCCP treatment was terminated by washing cells with fresh BG-11 media. (D) Growth curves of UV-C irradiated cells. Hours after medium exchange are shown on the horizontal axis, absorbance at 730 nm of cell culture is shown on the vertical axis as an index of cell growth ($n = 3$). Error bars represent standard deviation. Left panel shows the results in negative control without CCCP treatment. Right panel shows the results in the presence of 10 μM CCCP. UV- and UV+ represent with and without UV exposure, respectively. (E) Effects of UV-C irradiation on the strain genetically modified to uptake glucose. A schematic representation of modified strain and its function are illustrated (upper). In the $P_{trc}$::glcP strain, the glucose transporter gene was expressed with a WT background. Each image represents the growth following the UV irradiation under the UV+D condition. The symbols "–" and "+" indicate the absence and presence of D-glucose, respectively. Both with and without D-glucose, experiments were performed in the presence of 1 mM IPTG. Representative data of three independent experiments are shown.

triggered more rapid glycogen degradation leading to ~45% within 6 h (Fig 5B). More importantly, $\Delta glgP$ strain did not show significant glycogen degradation during 6-h dark exposure, regardless of the time when the cells were exposed to darkness (Fig 5B). These results suggest that the higher UV-C resistance is accompanied by lower glycogen degradation activity, such as in the wild type strain at subjective dawn and in the *glgP* strain throughout the circadian cycle.

To further analyze a link between UV resistance and glycogen metabolism, additional mutant strains lacking either the *glgC* or *glgX* gene were also generated. The *glgC* and *glgX* genes encode a glycogen synthesis enzyme (glucose-1-phosphate adenylyl transferase) and

degradation enzyme (isoamylase), respectively (Fig 5A) [35]. Disruption of the *glgX* gene is expected to reduce glycogen consumption, similar to the phenotype of the *glgP* strain with lower UV-C resistance. However, UV-C resistance of the *glgX* strain did not significantly change under the UV+D condition (S8B Fig), suggesting a possibility that GlgP constitutes a major glycogen degradation pathway. This possibility is consistent with our glycogen degradation assay in Δ*glgP* strain (Fig 5B). Therefore, disruption of *glgX* alone may be insufficient to reduce glycogen consumption. A strain with doubly disruption of both *glgP* and *glgX* genes was generated, which had essentially the same UV-C resistance property as the *glgP* strain (S8B Fig), which further supports the primary function of GlgP in glycogen degradation. Nullification of *glgC* is known to inactivate glycogen synthesis and leads to the constitutive loss of glycogen accumulation throughout the day [33]. Nullification of *glgC* is reported to severely restrict growth under 12-h:12-h LD cycles even without UV irradiation [39]. Assuming that circadian alternation in glycogen degradation is important for rhythmic UV-C resistance, the loss of glycogen synthesis by disruption of the *glgC* gene would abolish the circadian control of UV-C resistance. Under the UV+D condition, UV-C resistance was largely reduced in the *glgC* strain (S8B and S8C Fig). While the UV-resistance rhythm was not completely abolished, the amplitude was much lower than the wild type strain (S8C and S8D Fig). It should be noted that *glgC* disruption is known to enhance the magnitude of phase shifts against dark pulses due to an altered metabolic state, especially in the dark, whereas the amplitude of circadian transcriptional cycle in LL is less affected [33]. These observations further support a link between glycogen metabolism and UV resistance under the UV+D condition.

## Inhibition of ATP synthesis in darkness recovered the attenuated UV-C resistance at subjective dusk

When UV resistance is reduced in the wild type strain (subjective dusk), glycogen degradation activity is elevated (Fig 5B and [36]). However, as discussed earlier, glycogen accumulation itself also fluctuates in a time-dependent manner, and it remains unclear whether the accumulation itself or the degradation activity affects UV resistance. Thus, we hypothesized that increased *de novo* energy production due to glycogen degradation via enhancing catabolic pathways would reduce UV-C resistance. We evaluated the effect of *de novo* energy production via glycogen degradation in the dark, especially ATP synthesis, on UV-C resistance under UV+D condition. As shown in Fig 5C, following UV irradiation at hour 12 in LL, we treated WT cells with carbonyl cyanide 3-chlorophenylhydrazone (CCCP) as an uncoupler to inhibit ATP synthesis in darkness (for details, see Materials and Methods). Cells without CCCP suppressed cell growth after UV+D treatment (Fig 5D), same as shown in S2D Fig. On the other hand, CCCP addition did not restrict growth regardless of the presence or absence of UV-C irradiation (Fig 5D). It should be noted in the absence of UV, transient CCCP administration under our experimental condition did not affect growth profile much, comparable to control cells without CCCP. In other word, growth of the UV irradiated cells was rescued by CCCP treatment (Fig 5D). These results are consistent with the hypothesis that *de novo* ATP synthesis in the dark after UV-C irradiation severely limited growth, and the circadian fluctuation in ATP availability via glycogen degradation would drive the rhythm in UV resistance. This is also consistent with the reduced energy charge in the *rpaA* strain when the cells are exposed to the dark [37], remaining higher UV resistance as discussed above. Although ATP production is highly active in the light, UV resistance does not appear to be low, as shown in the WT strain under UV+L conditions. We believe that the higher UV resistance in the light is due to the higher DNA damage repair activity of photolyase, which masks the effects of UV irradiation. The very high UV sensitivity in the photolyase deficient strain under UV+L condition (S5C

Fig) can be interpreted that this strain is highly sensitive to UV in the light when ATP production is highly active without being masked by the photolyase activity.

## Forced incorporation of glucose constitutively lowered UV-C resistance

In order to validate whether changes in further downstream catabolic pathways affect UV-C resistance, we used a strain that was genetically modified to uptake exogenous glucose directly into the cells, as reported previously [40]. It should be noted that we also considered another possibility to use strains overexpressing the *glgP* and *glgC* genes to activate glycogen catabolism. However, overexpression strains may cause unexpected side effects. More importantly, it is obscure whether overexpression of metabolic enzymes really activates the magnitude of targeted metabolic pathways, considering the amount of the carbon source itself may not change. On the other hand, it has been shown in the forced glucose uptake strain that glucose is incorporated into OPPP and the Emden-Meyerhof pathway (EMP) to be used as *bona fide* carbon source (Fig 5E, [41]). In this strain, the *glcP* gene, which encodes a glucose transporter derived from another cyanobacterium, was expressed by the *trc* promoter in the presence of IPTG. In the dark, the growth of *Synechococcus* cells is suppressed. By contrast, forced glucose uptake enables the growth of *glcP*-expressing cells even in the dark [40]. The effect of forced activation of the glucose catabolism pathway on the circadian rhythm of UV-C resistance rhythm with dark exposure was investigated using the mutant and WT strains. As shown in Fig 5E, the addition of glucose and IPTG constitutively lowered UV-C resistance of the glucose uptake strain, whereas there was no change in the absence of glucose. UV-C resistance of the WT strain did not change regardless of the presence or absence of glucose (Fig 5E).

## Discussion

The results of this demonstrated that prokaryotic cyanobacteria also exhibit diurnal (Fig 1C) and circadian (Fig 2C and 2D) variations in UV resistance, as observed in eukaryotic algae. The necessity to adopt such a unique experimental (UV+D) condition is a possible reason why circadian control of UV resistance has not yet been reported in *Synechococcus*, which is among the most studied model organisms in chronobiology. In *Synechococcus*, the presence of darkness after UV irradiation significantly changed its time dependent UV sensitivity. It is similar to the daily changes in UV-B sensitivity observed in *Arabidopsis thaliana* [42]. In *Arabidopsis*, UV-B sensitivity is high in subjective dusk only when exposed to darkness after UV-B irradiation. It should also be noted that there is difference in experimental schedule between the two studies: in *Arabidopsis*, pre-UV irradiation with lower light intensity was applied a few hours before the UV-B irradiation, while such pre-irradiation was not necessary for our experiments. Nevertheless, the similarity suggests a common property in the circadian control of UV resistance across species.

Moreover, we performed genetic analysis to elucidate underlying molecular mechanisms, which strongly supported that UV resistance fluctuates depending on the state of glycogen-related metabolism. These observations in *Synechococcus*, a well characterized chronobiological model organism, may help to elucidate molecular mechanisms of circadian UV resistance control in other species, such as *Chlamydomonas* and *Euglena*. Under a 12-h:12-h LD cycle, UV resistance was greater in the Δ*kaiABC* strain than in the WT strain at ZT 18 (in Fig 1C). Considering the results under the UV+D condition, UV resistance is related to glycogen metabolism from ZT 12 to 24. It has been reported that total glycogen consumption in WT and the *kaiC*-null strain in the 12h of darkness is almost the same, but the consumption speed is slightly faster in the *kaiC*-null strain than in the WT strain [36]. Thus, the Δ*kaiABC* strain would consume more glycogen early in the dark period (ZT 12 to 18), and degrade less

glycogen during the second half of the dark period (ZT 18 to 24), compared with the WT strain. Such different metabolic kinetics between the WT and Δ*kaiABC* strains might be responsible for the difference in UV resistance at ZT 18.

The observations using CCCP treatment and forced incorporation of glucose (Fig 5) are consistent with our hypothesis that increased catabolic flux (from glycogen degradation to the downstream OPPP and glycolytic pathways) at subjective dusk causes lowered UV resistance, although further analysis is necessary to reveal the molecular mechanisms underlying the metabolism-gated, circadian UV-resistance system. It should be noted that in *Synechococcus* the rate of DNA synthesis is not under circadian control in LL but oscillates under LD condition [43]. This is because DNA replication of cyanobacteria stops after cells are exposed to the dark. However, the DNA replication activity immediately after dark exposure changes due to glycogen degradation activities in the dark. When glycogen degradation activity is relatively higher after cells are transferred to the dark, DNA replication persists with a longer duration due to elevated abundance of ATP-bound DnaA [44,45]. Since glycogen degradation activity of *Synechococcus* seems to increase at dusk, DNA replication continues for a while even after the onset of dark acclimation. If this is the case, UV irradiation and transfer to darkness (UV +D condition) at dusk, but not at dawn, would result in unstable replication of damaged DNA without photolyase-mediated repair. In *E. coli*, it is known that double strand breaks occur when the replication fork hits unrepaired DNA, resulting in lethality [46].

UV resistance is decreased during the time that more energy is generated by glycogen degradation. Recently, several lines of evidence on the relationship between glycogen metabolism and the circadian system have been reported in cyanobacteria [33,36,37,47,48]. These studies suggest that the circadian clock is necessary for the time-dependent accumulation and degradation of glycogen, which likely improves adaptive fitness under LD cycles [36,37,47]. Considering the allocation of energy resources, constitutively higher energy production *via* glycogen degradation throughout the day and night would be disadvantageous. Therefore, our observations should provide a better understanding of the adaptive strategy of cyanobacteria under LD cycles through the circadian system. Table 1 summarizes the UV-resistance profile of each strain used in this research and previously reported growth phenotypes under LD conditions. It appears that the growth of mutants with enhanced UV-resistance activity is restricted and vice versa, suggesting a trade-off between UV resistance and energy production *via* glycogen degradation. Trade-off between energy production and stress resistance (i.e. UV, oxidative and heat stress) is widely observed in various organisms, including *E. coli* [49,50] and *Saccharomyces cerevisiae* [51]. Thus, it seems plausible that this trade-off is controlled by a time-dependent metabolic state *via* the circadian oscillator, and is adjusted to an appropriate balance over a period of 24 h (Fig 6). Furthermore, this possible trade-off mechanism would answer the simple question of how merely keeping constitutively high level of UV resistance throughout the day and night is not observed in many organisms: in order to keep high UV resistance at all times even in the night would antagonize energy production processes when the carbon sources are limited, which in turn lowers adaptive fitness. Thus, the results of this

**Table 1. Summary of UV resistance properties and growth of each strain under the LD condition.**

|  | UV resistance (in this study) | Relative growth under the LD condition vs. the WT strain | References |
|---|---|---|---|
| WT | High at dawn, low at dusk |  |  |
| Δ*glgP* | Constitutively high | Defected | [47] |
| Δ*sasA* | Constitutively high | Defected | [15] |
| *cikA*⁻ | Constitutively low | Improved | [47] |
| P*trc*::*glcP* (+ glucose) | Constitutively low | Improved, grew also in the dark | [40] |

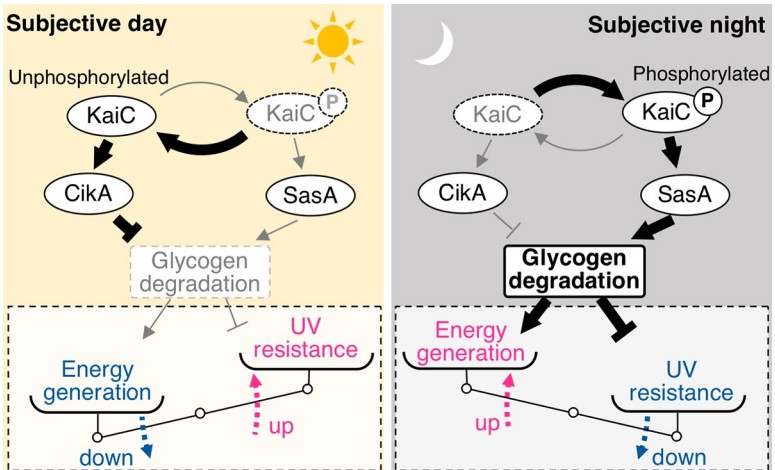

**Fig 6. Schematic diagram of a possible trade-off between UV resistance and energy metabolism in *Synechococcus*.** The results of the present and previous studies show an inverse correlation between UV resistance and glycogen degradation. High UV resistance at any time of the day would require continuous suppression of glycogen degradation, which must be disadvantageous to energy utilization in order to survive in the dark. Such a trade-off would give rise to the necessary use of different physiological functions in a time-dependent manner. Priority would be given to UV resistance in the daytime when UV irradiation is present, whereas energy production *via* glycogen degradation in the night occurs when photosynthesis is interrupted. Such a time-dependent trade-off is plausibly controlled by the output pathways mediated by the SasA and CikA from the Kai-based central oscillator.

study provide the circadian system as a well-controlled adaptation system that appropriately allocates resources and optimizes the trade-off among physiological processes in a time-dependent manner.

## Materials and methods

### Bacterial strains and culture conditions

All strains were modified from the pseudo-wild-type (WT) *S. elongatus* PCC 7942, designated as strain NUC42 [9], which contained the P*kaiBC*::*luxAB* bioluminescence reporter segment at a targeting site NS I. The derivatives used in this study are listed in S1 Table. To construct plasmids to disrupt the *Synechococcus* genes *phr*, *glgP*, *glgC*, *glgX*, and *uvrA*, the 500-bp upstream region of each target gene, an antibiotic resistance gene, and the 500-bp downstream region of each target gene were amplified and assembled by polymerase chain reaction (PCR), then cloned into the pGEM-T easy vector (Promega Corporation, Madison, WI, USA). The previously constructed plasmids pDkaiABC, pAM2176, and pAM2152 were used to disrupt the clock-related genes *kaiABC*, *sasA*, and *cikA*, respectively [9,52]. To yield short-period mutant strains and its negative control (rescued *kaiC*$^{WT}$), we re-introduced *kaiABC* and mutated *kaiABC* genes into *kaiABC*-null background by previously constructed plasmids and its derivative (pCkaiABC [9], pIL788 (equivalent to pCkaiABC$^{A87V}$) and pIL782 (equivalent to pCkaiABC$^{F470Y}$)). pIL788 and pIL782 were constructed by introducing mutated *kaiC* DNA fragment into *Eco*RV-*Bsp*EI sites of pCkaiABC. To yield strains expressing *phr* from the *trc* promoter at a targeting site NS II, a DNA fragment containing the *phr* open reading frame was amplified by PCR using WT *Synechococcus* genomic DNA as a template and then cloned into pNS2KmTΔHincII-Ptrc [53]. pAL46, a plasmid for constructing the glucose uptake strain, was provided by Prof. Shota Atsumi of the University of California, Davis (Davis, CA, USA). The *glcP* gene expressed by the *trc* promoter was introduced into the NS I site using the plasmid pAL46. Detailed information of pAL46 was described previously [40]. All strains were

generated by natural transformation with plasmid DNA [54]. Cells were grown at 30˚C with illumination of ~40 μmol/m$^2$·s on either BG-11 liquid or solid (containing 1.5% agar) media supplemented with appropriate antibiotics.

## Bioluminescence assay

The P$_{kaiBC}$::$luxAB$ bioluminescence profile was monitored on solid media with photomultiplier tubes under continuous light conditions after synchronization of the cells to a 12-h:12-h LD) cycle, as described previously [55].

## Evaluation of UV resistance by spot assays to assess growth

In order to evaluate UV resistance, each *Synechococcus* strain was cultured in BG-11 liquid media, diluted to an optical density at 730 nm (OD$_{730}$) of 0.2. The aliquots were further diluted with BG-11 media to 1:100, and 2 μL of each dilution was spotted onto solid BG-11 media without antibiotics. Cells were synchronized to twice 12-h:12-h light:dark (LD) cycles, and then released into each experimental condition as described in the text. UV exposure was initiated just before the dark/light transitions under diurnal condition (ZT = 0, 12, 24) and UV+D condition. UV-C irradiation was performed using a UV Crosslinker (model, CL-1000; Analytic Jena AG, Jena, Germany) equipped with a discharge tube with a main wavelength of 254 nm. The irradiation intensity was set at 500 J/m$^2$. UV-B irradiation was performed using UV transilluminator (PI-20.AB; BIO CRAFT, Tokyo, Japan) equipped with a discharge tube with a main wavelength of 312 nm. UV-B intensity was measured by a digital UV radiometer (Radiometer Sensor, UVX-31; Analytic Jena AG, Jena, Germany). The UV-irradiated cells on agar plates were exposed to darkness and then to light for the "UV+D" condition experiments or cultured without dark exposure in the light for the "UV+L" experiments. Cellular growth was evaluated by densitometric analysis of each spot captured using a single-lens reflex camera (Pentax model K20D; Ricoh Company, Ltd., Tokyo, Japan). Quantification of UV resistance (growth index) was performed using ImageJ (https://imagej.nih.gov/ij/) as described in [56]. UV resistance of each strain was normalized to the densitometric value of a corresponding negative control strain without UV exposure. The quantified densitometric value (*n* = 3) were obtained from experiments performed at the same time using three independently pre-cultured cells. We have confirmed that at least two or more completely independent experiments (independently precultured cells to be analyzed on different plates at different time) obtained essentially similar results.

## Evaluation of UV resistance and effect of ATP synthesis inhibitor by monitoring growth in liquid cultures

In order to evaluate UV resistance in liquid culture, each *Synechococcus* strain was cultured in BG-11 liquid media, diluted to an optical density at 730 nm (OD$_{730}$) of 0.075. Cells in liquid media were entrained to two 12-h:12-h light:dark (LD) cycles in 6-well or 24-well plastic plates with shaking, and then released into each experimental condition as described in the text. UV-C irradiation was performed using a UV Crosslinker as described above in spot assays and the irradiation intensity was set at 500 J/m$^2$. For evaluating the growth of cells, the absorbance at 730 nm was measured by microplate reader (Powerscan HT; DS Pharma Biomedical, Ltd., Osaka, Japan). In order to evaluate the effects of blocking ATP synthesis, CCCP (Carbonyl cyanide 3-chlorophenylhydrazone, C2759, Sigma-Aldrich) was added to the cultures at a final concentration of 10 μM following UV irradiation at hour 0 or 12 in LL conditions. For negative control, the same volume of ethanol instead of CCCP was added. Cultures were exposed

to darkness for 6 h and CCCP treatment was terminated by twice washing of cells with fresh BG-11 media.

## Microscopy and image acquisition

UV-irradiated cells in liquid cultures under UV+D condition were prepared as described above. UV-irradiated cells were mounted between cover glass and imaged with a confocal laser scanning microscope (FV1000; Olympus, Japan) equipped with 40x objective lens. Image acquisition were done on FV10-ASW software applied to this system. Autofluorescent images of *Synechococcus* cells were collected after 559 nm excitation using a 550–600 nm emission window. Differential interference contrast images were obtain using TD1 channel in FV10-ASW.

## DNA repair activity detection with a cyclobutane pyrimidine dimer (CPD)-specific enzyme-linked immunosorbent assay (ELISA)

Briefly, *Synechococcus* cells at an $OD_{730}$ of ~0.3 were grown in BG-11 liquid media in a continuous culture system in a 1.5-L flat-bottom glass bottle at 30˚C with irradiation of 40 μmol/m$^2$·s generated by white fluorescent lamps. Cells were sampled at hour 0 or 12 in LL, irradiated with UV-C, as described above, and then harvested and stored at –80˚C until assayed. Genomic DNA was extracted from the harvested cells using the phenol–chloroform extraction method and Cica Geneus DNA Prep Kit (for plants; Kanto Kagaku, Tokyo, Japan). CPD contained in the genomic DNA was detected and quantified using the High-Sensitivity CPD ELISA Kit (Cosmo Bio Co., Ltd., Tokyo, Japan) in accordance with the manufacturer's protocol.

## Screening and identification of UV-resistant mutants

WT *Synechococcus* was transformed with a genomic DNA library fused to a Tn5-derived transposon with the kanamycin resistance gene [57] in order to introduce random mutations. This genomic DNA library was provided by Dr. Mitsunori Katayama of Nihon University (Tokyo, Japan). Transformants were streaked onto solid BG-11 plates containing kanamycin. After synchronizing the cells to two 12-h:12-h LD cycles, transformant cells were exposed to the light for 12 h, irradiated with UV-C as mentioned above, exposed to the dark for 6 h, and then returned to the light. Several days after UV irradiation, among 5,000 transformant clones screened, 18 colonies formed on the plates were initially picked up as possible mutants with greater UV resistance. Four out of the 18 clones were confirmed to show elevated survival rates after the UV+D treatment at ZT 12 by spot tests. All the four clones are reported in the present paper. Genomic DNA was extracted from each of the four mutant strains, and the transposon insertion sites were identified by inverse PCR and sequence analysis.

## Evaluation of H$_2$O$_2$ resistance

In order to evaluate $H_2O_2$ resistance, WT *Synechococcus* cells were cultured in liquid BG-11 media and diluted to an $OD_{730}$ of 0.2. Cells in liquid media were entrained to two 12-h:12-h light:dark (LD) cycles in 24-well plastic plates with shaking, and $H_2O_2$ was added to the cultures at a final concentration of 0.01–10 mM at hour 0 or 12 in LL conditions. Following the addition of $H_2O_2$, each cell suspension was exposed to light or dark for 6 h to imitate the UV +L or UV+D condition. Then, 2 μL aliquot was spotted onto solid media and cultured under LL conditions. The experimental schedule is shown in S6A Fig. Cellular growth was evaluated by densitometric analysis of each spot.

## Quantification of glycogen content

*Synechococcus* cells at an $OD_{730}$ of ~0.3 were grown in BG-11 liquid media at 30°C with air bubbling and irradiation with 40 μmol/m²·s of white fluorescent lamps, and were entrained to two 12-h:12-h light:dark (LD) cycles. At hour 0 or 12 in LL, cells were exposed to darkness and harvested at hour 0, 1, 3 and 6 in darkness for extracting and quantifying the glycogen in the cells. Harvested cells were resuspended with sterilized water and disrupted with 0.1 mm glass beads using a Multi-beads Shocker (Yasui Kikai, Osaka, Japan). The homogenates were boiled for 10 min to inactive the endogenous enzymes, and then insoluble fraction was removed by centrifugation. Glycogen content of supernatant was measured using the Glycogen Colorimetric/Fluorometric Assay Kit (BioVision, Inc., US) in accordance with the manufacturer's protocol. For colorimetric assay, the absorbance at 570 nm was measured with a microplate reader (Powerscan HT; DS Pharma Biomedical, Ltd., Osaka, Japan). Glycogen contents were determined using a standard curve and normalized by $OD_{730}$ units.

## Supporting information

**S1 Fig. The effects of UV-C irradiation under the UV+L condition.** (A) UV-dose dependency under the UV+L condition. Growth of the UV-C-irradiated cells. Each image represents a spot assay of cellular growth after UV irradiation of 0–1000 J/m² at hour 12 or 24 in the light (L12 or L24) under the UV+L condition (as shown in Fig 1D). Representative data of three independent experiments are shown. (B) Densitometric analysis of the growth test in S1A Fig. The intensity of UV irradiation is shown on the horizontal axis, and relative UV resistance is shown on the vertical axis, as in Fig 1C (n = 3). Error bars indicate the standard deviation. (C) Phase responses against UV-C Irradiation. A schematic representation of the experimental schedule. After the cells were synchronized to two 12-h:12-h LD cycles, bioluminescence rhythms were monitored to measure the *kaiBC* promoter activity under LL condition. Cells were subjected to UV-C irradiation at each time point (arrowheads). (D) Bioluminescence rhythms of cells subjected to UV-C irradiation at the indicated time points. The results indicate the UV-C irradiation did not affect the phase of the endogenous oscillator under our experimental conditions.
(TIF)

**S2 Fig. Time-dependent variation of growth under UV+ D condition was not caused by darkness alone.** (A) Growth of the cells under UV+D condition or "6-h darkness only" condition without UV-C irradiation (WT, wild-type; Δ*kaiABC*, *kaiABC*-deficient strain). Spot tests were performed with cell suspension with an optical density at 730 nm (OD730) of 0.2 and 0.002. (B) Densitometric analysis of the growth test in S2A Fig (using the results for spots starting from cell suspension with OD730 of 0.2). The value for dark-exposure-only sample at hour 0 for each strain was normalized to the value of 1.
(TIF)

**S3 Fig. Circadian variation of survival rates and growth under the UV+D condition.** (A) Circadian variation of survival rates after UV-C irradiation under the UV+D condition. A schematic representation of experiment. WT cells were cultured and synchronized to a 12-h:12-h light:dark (LD) cycles in BG-11 liquid media, then plated on agar media at each time point, and subjected to UV irradiation under the UV+D condition. The plates were further incubated under continuous light. (B and C) The survival rates of UV-C irradiated WT cells under the UV+D condition. Survival rates were calculated by counting the colony-forming units and normalized to UV(−) control samples. (D) Time dependent reaction to UV Irradiation in liquid culture. Growth of the UV-C irradiated WT cells under the UV+D condition. UV irradiation was performed at hour 0 or 12 in LL. The growth curve in each condition was shown (left panel, L0; right panel, L12). Time after

UV irradiation and subsequent darkness are shown on the horizontal axis. $OD_{730}$ of cell culture is shown on the vertical axis (UV–; negative control without UV irradiation). (E) Images of UV irradiated cells. Cells were sampled from the experiments shown in S3D Fig at 60 h after UV+D treatment. Images show an overlay of differential Interference contrast (gray) and autofluorescent (red) microscopic pictures. Autofluorescence signals of cells due to photosystems were obtained (for details, see Materials and Methods).
(TIF)

**S4 Fig. Results of changing each parameter under the UV+D condition.** (A) Time-dependent response to UV-B irradiation. Growth of the UV-B irradiated cells. UV irradiation was performed using a discharge tube with a main wavelength of 312 nm (UV-B intensity was 5.22~5.88 mW/cm$^2$). Each photo represents spot-plate growth following UV-B irradiation. Left label represents the time of UV irradiation and upper label represents durations of UV-B exposure. (B) Densitometric analysis of the growth test in S4A Fig. UV resistance was normalized to the densitometric value of a corresponding negative control strain without UV exposure. Error bars indicate the standard deviation. (C) The effects of changing light periods in response to UV-C irradiation at hour 12 in the light. A schematic representation of the experimental schedule is shown on the left, and the growth of the UV-C-irradiated cells is shown on the right. Each schedule on the left side is arranged to correspond to the experimental results. Representative data of three independent experiments are shown. (D) Effects of changing dark periods after UV irradiation at hour 12 in the light under the UV+D condition. Each symbol or image is the same as in S4C Fig.
(TIF)

**S5 Fig. UV-C resistance of several DNA repair mutants.** (A and B) A schematic representation of the experimental schedule. Each symbol is the same as in Figs 1D and 2A. (C) UV-C resistance of the Δ*phr* strain under the UV+L condition. Spot assay to assess growth of the UV-C-irradiated cells. UV irradiation was administrated at each indicated time under the UV +L condition. (D and E) Effect of the addition of IPTG to Δ*phr*; P*trc*::*phr* strain under the UV +D condition. S5D Fig and Fig 3A are based on the same experimental design, while the images were obtained from independent experiment as the IPTG- negative control against the IPTG+ data shown in S5E Fig. Each image represents a spot assay to assess growth under the UV+D condition as shown in Fig 2B. Experiments were performed in the absence of IPTG (D) or the presence of 100 μM IPTG (E). (F and G) UV-C resistance of the Δ*uvrA* mutant under each experimental condition. Growth of UV-C-irradiated cells (WT; Δ*uvrA*, *uvrA*-deficient strain). Each image represents a spot assay to assess growth following UV irradiation at each time point under the UV+L condition (F) and the UV+D condition (G). (H and I) Densitometric analysis of the growth test in S5F Fig (H) and S5G Fig (I). Since colony formation was severely suppressed in the *uvrA* strain, it was difficult to obtain appropriate values when applying the usual densitometry analysis due to the effect of background noise. Therefore, the area of the colony was shown as a dashed red line to validate its rhythmicity. Values of the colony area differed significantly between hours 0 and 12 and between hours 12 and 24 (** $P< 0.01$, Student's *t*-test), supporting the rhythmicity. (J) *uvrA* mRNA accumulation profile in each mutant strain. The *uvrA* expression level from microarray data reported in the previous studies is shown. The microarray data were extracted for ΔkaiABC and the corresponding control wild type (WT) strains from Ito et al. [14].
(TIF)

**S6 Fig. $H_2O_2$ resistance of WT strain at L0h and L12h.** (A) A schematic representation of the experimental schedule. After the cells were synchronized to two 12-h:12-h LD cycles, $H_2O_2$

was added to the liquid cultures at hour 0 or 12 in LL, respectively (arrowhead). Following the addition of $H_2O_2$, each cell suspension was exposed to light or dark for 6 h, then spotted onto solid media and cultured under the LL condition. (B) Each image represents a spot assay to assess growth following the addition of $H_2O_2$. Upper panels represent the final concentration of $H_2O_2$ ((−) control; negative control without $H_2O_2$). Light and dark on the left side represent conditions after the addition of $H_2O_2$.
(TIF)

**S7 Fig. Circadian bioluminescence rhythms were abolished in Tn-5-inserted *mutants 2* and *3*.** (A) Detailed sequences of Tn-5 insertion sites shown in Fig 4B. The intermediate sequence of the Tn-5 transposon was shown with asterisk (*) and only both ends of the transposon sequence are shown in yellow. The numbers indicate the positions of insertion sites when the initial nucleotide for the translational start codon of each ORF is assigned a positional value of 1. (B) Bioluminescence rhythms of WT, *mutant 2* and *mutant 3* cells that carried the $P_{kaiBC}$:: *luxAB* reporter cassette. The cells were grown on solid media under the LL condition after two LD cycles at 30 μmol/m$^2$·s and bioluminescence were measured using photomultiplier tubes ($n = 3$). (C) For clarity, the bioluminescence profiles of *mutants 2* and *3* are also shown in magnified scales (magnified scale shown on the right).
(TIF)

**S8 Fig. Relationships between UV-C resistance and glycogen metabolisms.** (A) Glycogen contents fluctuate in a circadian fashion under LL condition. Glycogen contents in WT were quantified. Hours in LL is shown on the horizontal axis, and glycogen content is shown on the vertical axis. The glycogen contents are normalized to the $OD_{730}$ unit of harvested cells. (B) Effects of UV-C irradiation on glycogen-related mutants under the UV+D condition (Δ*glgP*, *glgP*-null mutant; Δ*glgX*, *glgX*-null mutant; Δ*glgC*, *glgC*-null mutant). Each image represents a spot assay to assess growth as shown in Fig 2B. It was difficult to evaluate UV-C resistance of the *glgC* strain due to dramatically lower viability under a 12-h:12-h LD cycle. Therefore, experiments were carried out with spots with ten-fold denser cell suspensions (Δ*glgC* and WT at the bottom). It should be noted that UV-irradiation at hour more severely suppressed colony formation in the WT cells than the *glgC* strain (red square), suggesting not only the magnitude of UV resistance but amplitude of the rhythm was greatly reduced in the mutant strain. (C) Densitometric analysis of the growth test in S8B Fig (Δ*glgC* and WT, spots with ten-fold denser cell suspensions). The timing of UV irradiation is shown on the horizontal axis, and relative UV resistance is shown on the vertical axis, as in Fig 1C (n = 3). Error bars represent standard deviation. (D) Trough to peak ratio of the UV resistance profiles in the wild type and *glgC* strains, using the maximal and minimal values of the UC resistance profiles shown in S8C Fig. The ratio significantly differs between the two species. **$P < 0.01$ (Student's t-test).
(TIF)

**S1 Table. Strains used in this study.**
(DOCX)

**S1 Dataset. The source data.** The source data underlying Figs 1–6 and S1–S8 Figs are provided.
(XLSX)

## Acknowledgments

We would like to thank Dr. Mitsunori Katayama (Nihon University) for providing the transposon plasmid library and Prof. Shota Atsumi (University of California Davis) for providing

plasmids to construct the glucose uptake strain. We also thank the members of the Iwasaki lab for valuable discussions.

## Author Contributions

**Conceptualization:** Koji Kawasaki, Hideo Iwasaki.

**Data curation:** Koji Kawasaki, Hideo Iwasaki.

**Formal analysis:** Koji Kawasaki.

**Funding acquisition:** Koji Kawasaki, Hideo Iwasaki.

**Investigation:** Koji Kawasaki, Hideo Iwasaki.

**Methodology:** Koji Kawasaki.

**Project administration:** Koji Kawasaki, Hideo Iwasaki.

**Resources:** Koji Kawasaki, Hideo Iwasaki.

**Supervision:** Hideo Iwasaki.

**Validation:** Hideo Iwasaki.

**Visualization:** Koji Kawasaki.

**Writing – original draft:** Koji Kawasaki, Hideo Iwasaki.

**Writing – review & editing:** Koji Kawasaki, Hideo Iwasaki.

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
