## [Decision Letter · Decision Letter 0]

4 Aug 2020

Dear Dr Iwasaki,

Thank you very much for submitting your Research Article entitled 'Involvement of glycogen metabolism in circadian control of UV resistance in cyanobacteria' to PLOS Genetics and apologies for the slow review. Your manuscript has been evaluated by three prestigious experts in the field.  As you will see below, the reviewers praise the work but also raise concerns about the current manuscript. Based on the reviews, we will not be able to accept this version of the manuscript, but we would be willing to look at a revised version. We cannot, of course, promise publication at that time.

Should you decide to revise the manuscript for further consideration here, your revisions should address the specific points made by each reviewer. We will also require a detailed list of your responses to the review comments and a description of the changes you have made in the manuscript. Please aim to resubmit within the next 60 days, unless it will take extra time to address the concerns of the reviewers, in which case we would appreciate an expected resubmission date by email to plosgenetics@plos.org.

[LINK]

Please do not hesitate to contact us if you have any concerns or questions.

Yours sincerely,

Josep Casadesús

Section Editor: Prokaryotic Genetics

PLOS Genetics

Reviewer's Responses to Questions

**Comments to the Authors:**

Reviewer #1: The paper by Kawasaki and Iwasaki reports a study of a circadian-clock mediated response to UV light in the unicellular cyanobacterium Synechococcus elongatus. The authors found expression of an activity of resistance to UV light dependent on central clock components (Kai proteins). Importantly, the resistance is hidden by the very active DNA photolyase of cyanobacteria, which made the authors to carry out an exhaustive genetic analysis to be able to demonstrate the phenomenon. The expression of resistance is higher at dawn, preparing the organism to fight UV during daytime. This is an important finding. The authors then carried out transposon mutagenesis and identified genes participating in the circadian UV-resistance response. They found circadian clock-related genes, confirming the original finding, and mutations in gene glgP encoding a glycogen degradation enzyme. Complementary experiments allow the authors to propose that UV-resistance is counteracted by active metabolism through an unknown mechanism. The paper is well written and the experiments described are sound. I have only a few formal points:

1. The last paragraph of the introduction is very reiterative with the abstract. You may reduce it significantly.

2. Lines 121-122: It is hard to evaluate how significant this difference is.

3. Line 149: 60 hours (“hours” missing).

4. Line 159: please indicate the time units for UV-B irradiation in Fig. S3A.

5. Line 197: “The photorepair reaction is…”, not “are”.

6. Line 199: please spell out CPD.

Reviewer #2: The manuscript, “Involvement of glycogen metabolism in circadian control of UV resistance in cyanobacteria” by Kawasaki and Iwasaki, is a worthy addition to the literature on circadian rhythms in cyanobacteria, with important implications for daily rhythms of resistance to UV light that have been hypothesized to underlie the early evolution of circadian rhythmicity. There are a number of issues that the authors must address in a revised manuscript:

1. Abstract (line 49): remove the phrase “through unknown mechanism(s).” These three words make the Abstract sound weak and they are unnecessary because this issue is thoroughly addressed in the DISCUSSION.

2. Fig. 1B/E and C/F: given that the LD and LL protocols are the same for the first 12 hours after the beginning of the UV exposures, the UV resistances SHOULD be the same for hours 0 & 6 (and 12 ?, see next point) in panels B/C versus panels E/F. Why aren’t they?

3. Fig. 1B/C: for the “12” and “24” hour timepoints, was the UV exposure initiated just before or just after the dark/light transitions?

4. Fig. 1 legend: what does “three independent experiments” mean? Does it mean three experiments with an n of 1? Or three experiments with multiple replicates each? Or do the authors actually mean three triplicates within one experiment performed at the same time?

5. Fig. 2A/B/C: is the UV- control a culture that is maintained in LL the entire time? From the perspective of optimal experimental design, there should be UV- controls that experienced the 6h dark exposures at each timepoint. If that experiment has not been done, the authors need to do it.

6. Fig. 2D: the test with short period mutants is excellent, but the conclusion would be strengthened if the authors also show equivalent data for a long-period mutant.

7. Fig. S2: the data for liquid culture only covers 1 cycle in LL, while for the spotted cultures, the researchers tested 2 cycles (Fig. 2). It would be good to do 2 cycles in the liquid cultures also to confirm a self-sustained rhythm.

8. Lines 164-165: which figure are the authors referring to ?

9. Line 197: "are" should be "is"

10. Page 8: define “CPD” (Cyclobutane Pyrimidine Dimer). I had to look it up because this abbreviation is not defined in the manuscript.

11. Fig. 3B: show histograms for the UV- control data. The way the data are currently presented imply that there was no control.

12. Fig. S4F/G: From the authors’ previous microarray studies, was UVR rhythmically expressed? I’m not clear on the authors’ logic here: if the deletion of UVR causes the loss of rhythmic resistance in the UV+D protocol (Fig. S4G), isn’t that result consistent with rhythmic regulation of UVR being responsible for the results shown in Fig. 2 ?

13. Fig. 4: please give specific information as to the location and identity of Mutants 1-4 (e.g., provide insertion sites and the transposon; perhaps adding to the Supplement the sequence of the transposon with 50 nucleotides of the adjacent gene sequences on either side of the transposon sequence would suffice for each Mutant). Other researchers may want to repeat your results.

14. Fig. 4: Puszynska and O’Shea reported (2017 eLife) that ΔrpaA has a lower glycogen content compared with WT, but why didn’t Mutant 3 in rpaA (Fig. 4A) show a lower resistance to UV?

15. Fig. 5: Has the glgC gene been overexpressed to assess whether the glycogen content is altered and its effect on UV resistance?

16. Line 383: “contrast” is misspelled.

Reviewer #3: This is a very interesting manuscript that links UV sensitivity to oxidative phosphorylation in cyanobacteria.

The authors characterize the circadian control of UV sensitivity in the cyanobacterium Synechococcus elongatus PCC. They demonstrate that the UV sensitivity changes throughout the day with a maximum at dusk under light/dark cycles. They make the observation that when cells are maintained under constant light there are not cyclic changes in UV sensitivity, but that a dark period after the UV treatment reveals cyclic sensitivity. Using mutants of DNA repair enzymes, they demonstrate that the cyclic sensitivity does not seem to be strongly regulated by either the photolyase or enzymes involved in nucleotide excision repair. The authors argue that the ‘necessary’ dark period after the UV treatment inhibits the photolyase activity revealing the cyclic sensitivity under constant light conditions. The authors also screened a mutant population for lines with high resistance to UV at dusk. They discovered mutations in circadian clock components that lock gene expression in a ‘dawn’ state, which has an associated high UV resistance. In addition, they identified two lines with mutations in a glycogen-phosphorylase gene. Further experiments demonstrate that consumption of glucose via oxidative phosphorylation in the dark is associated with high UV sensitivity.

The manuscript is clear and well written the results presented will be relevant to researchers working on UV sensitivity in all photosynthetic organisms, as well as researchers working on the role of mitochondria on oxidative stress. My main issues with this manuscript are:

1. I think their results link UV to mitochondrial oxidative phosphorylation, and glycogen metabolism is just an example of a more general principle. I elaborate on this point below.

2. Similar results of the ‘dark’ requirement for circadian controlled sensitivity were described in land plants by Takeuchi et al.(2014) (doi:10.1093/jxb/eru339). Although the authors cite the work of Feher et al, 2011, these authors only report UV mediated control of gene expression and not sensitivity, which did not appear to be circadian controlled in constant light, as observed in cyanobacteria. Takeuchi et al.(2014) demonstrated circadian controlled UV sensitivity that is darkness dependent in the model plant Arabidopsis thaliana, carrying experiments analogous to the ones reported here. This does not minimize the importance of the current study, on the contrary, it shows that the phenotype and mechanism reported might be widely relevant in photosynthetic organisms.

3. There are a couple of instances in which I don’t agree with the written description of the results. These discrepancies don’t have a big impact on the main conclusions of the paper but should be corrected. More details are below.

(1)The link with oxidative phosphorylation: I believe the authors should change the title to ‘Involvement of mitochrondrial oxidative phosphorylation in circadian control of UV resistance in cyanobacteria’. The authors show that the availability of carbon for respiration is linked to increased sensitivity to UV at dusk. This effect is not necessarily glycogen dependent, since feeding the cyanobacteria glucose in the dark has the same effect. In addition, the authors demonstrate that the inhibition of oxidative phosphorylation suppresses UV sensitivity at dusk. Under constant light, oxidative phosphorylation is not critical for the cells since they are photosynthesizing. It seems that the correlation of oxidative phosphorylation with DNA synthesis leads to high sensitivity to UV. Since the bacteria cells expressing the glucose transporter can be grown under constant darkness in the presence of glucose, the authors could further test whether the circadian control occurs in constant darkness. Are rhythms of oxygen consumption in the dark correlated with changes in UV sensitivity?

In addition, I don’t fully understand the relationship between SasA, RpaA, glgP and glycogen content. The loss of SasA leads to low glgP expression (Fig. 4). The loss of glgP leads to a decrease of glycogen degradation in the dark. But why is the glycogen content in the SasA mutant lower than the wild type at dusk as reported previously? Could the authors confirm that result?

The authors also need to be more careful and precise when discussing the role of metabolism. For example, on line 364 they appear to argue that ‘de novo energy production via glycogen degradation in the dark’ is involved in the changes in sensitivity. The cells produce a lot of ATP during the light period without any apparent negative effect, so ATP production per se is not an issue, however mitochondrial oxidative phosphorylation appears to be the link to increased sensitivity. In the abstract and the discussion the authors also argue that glycogen degradation or ‘increased metabolic flux’ is required for the increased UV sensitivity. That is actually not the case as their nice experiments using the glucose feeding demonstrate. ‘Metabolic flux’ is an ambiguous term in this context.

In line 439 the authors argue that high UV resistance would antagonize energy production process. Not necessarily, since that is not a problem in the light. I would rather conclude that under carbon limitation in the dark a decrease in UV-resistance might provide a fitness advantage. However, it is also possible, that mitochondria might be more susceptible to UV stress when oxidative phosphorylation is high due to the production of reactive oxygen species.

(3)The written description of the results does not seem to match the figures in the following examples:

a. Line 255: ‘ inactivation of uvrA gene, […], had not effect on UV-C sensitivity under the UV+L condition’. Fig. S4F demonstrates that this mutant was more sensitive than the wild type in UV+L and UV+D, although the effect was stronger in UV+D. In this and other cases, would be clearer to show the quantification of the cell density analysis with the necessary replicates and statistics instead of showing only the images.

b. The authors did not measure glycogen levels in the glgX mutant strain, they cannot conclude its effect on glycogen based on the effect of this mutation of UV sensitivity. The experiments shown in Fig. S7B prove the effect of glgP in UV-C sensitivity not in glycogen metabolism, since no glycogen was measured.

c. Line 353: the authors state that in the glgC mutant strain UV-C resistance was decreased constitutively (Fig. S7B), however the 10x experiments demonstrate that that is not the case, at least based on the figure shown. The cell/colony density should be quantified and the authors should carry out proper statistical analyses.

Other comments:

Figure S4C is mentioned before S4A and B in the text. The authors might want to change the order of the labels for clarification.

Is Figure 3A the same as Figure S4D?

The authors define the term zeitgeber (ZT), it would help using it throughout the text and figures to better unambiguously define the time. For example, line 273, the authors write ‘at hour 12’ which is ambiguous.

The authors should mention that DNA synthesis occurs at dusk in Synechococcus (Mori et all, 1996) in both LD and LL.

Line 331: the authors state that the half-life of glycogen is ~ 6h. However, the authors did not measure glycogen at dawn, only during the first 6 h after dusk, so they cannot estimate its half-life. It is expected based on other experiments to be 6h, or better said, half of the dark period since cells are likely to optimize glycogen degradation, so that they don’t run out of carbon before dawn in a similar manner as starch in land plants.

Line 361: I believe the authors mean ‘subjective dusk’ instead of ‘subjective dawn’.

Line 483, please provide the intensity of the UV-B light use.

More details about the mutant screen would be useful. How many cells were irradiated? How many initially ‘resistant’ colonies were found? After how many days did colonies form after irradiation?

**Have all data underlying the figures and results presented in the manuscript been provided?**

Reviewer #1: Yes

Reviewer #2: Yes

Reviewer #3: Yes

PLOS authors have the option to publish the peer review history of their article (what does this mean?). If published, this will include your full peer review and any attached files.

Reviewer #1: No

Reviewer #2: No

Reviewer #3: No

---

## [Decision Letter · Decision Letter 1]

24 Oct 2020

Dear Dr Iwasaki,

Thank you very much for submitting your Research Article entitled 'Involvement of glycogen metabolism in circadian control of UV resistance in cyanobacteria' to PLOS Genetics. Your manuscript was fully evaluated at the editorial level and by independent peer reviewers. The reviewers appreciated the attention to an important topic but identified some aspects of the manuscript that should be improved.

We therefore ask you to modify the manuscript according to the review recommendations before we can consider your manuscript for acceptance. Your revisions should address the specific points made by each reviewer.

[LINK]

Yours sincerely,

Josep Casadesús

Section Editor: Prokaryotic Genetics

PLOS Genetics

Reviewer's Responses to Questions

**Comments to the Authors:**

Reviewer #2: Please see attachment, "Comments to authors.pdf"

Reviewer #3: I first apologize for appear to indicate that cyanobacteria have mitochondria my previous comments to the authors, that is embarrassing. However, I think the authors understood my points which were related to oxidative phosphorylation and electron transport.

The authors have now addressed all my initial questions and concerns.

I have just one short comment on the new paragraph that addresses the role of rpa in glycogen metabolism.

The authors write:

“ … glycogen content was lower in the rpaA strain despite the fact that glycogen synthesis activity was kept high in the rpaA strain”.

I am not sure what the authors mean by ‘glycogen synthesis activity’. Puszynska and O’Shea measured the maximum catalytic activity of enzymes involved in the glycogen synthesis, not the actual flux towards starch, therefore I would change the wording to:

“ … glycogen content was lower in the rpaA strain despite the fact that the maximum catalytic activity of enzymes involved in glycogen biosynthesis are not reduced in the rpaA strain”.

The authors also write:

“When cells are arrested in the dawn phase, metabolic flows including carbon sources may not operate since resources for glycogen synthesis are depleted”.

I am unclear what the authors mean by ‘metabolic flows including carbon sources’ and by ‘resources for glycogen synthesis are depleted’. The author need to clarify this point. Puszynska and O’Shea show that the energy charge is low at night in the rpaA strain indicating that this strain cannot regulate its energy balance efficiently, reacting strongly to light off (and on) conditions. Changes in ATP metabolism in the rpaA strain fit with the authors’ results using CCCP. A strong decrease energy charge at the beginning of the night (Puszynska and O’Shea) correlates with higher resistance in the rpaA strain. A strong drop in energy charge when the light goes off would also be expected in other mutants lacking glycogen that maintain high UV resistance. Why energy balance would affect UV sensitivity is the remaining open question.

**Have all data underlying the figures and results presented in the manuscript been provided?**

Reviewer #2: Yes

Reviewer #3: Yes

PLOS authors have the option to publish the peer review history of their article (what does this mean?). If published, this will include your full peer review and any attached files.

Reviewer #2: No

Reviewer #3: No

---

## [Editor Report · Decision Letter 2]

28 Oct 2020

Dear Dr Iwasaki,

I am pleased to inform you that your manuscript entitled "Involvement of glycogen metabolism in circadian control of UV resistance in cyanobacteria" has been editorially accepted for publication in PLOS Genetics. Congratulations!

Yours sincerely,

Josep Casadesús

Section Editor: Prokaryotic Genetics

PLOS Genetics

Comments from the reviewers (if applicable):

**Data Deposition**

http://datadryad.org/submit?journalID=pgenetics&manu=PGENETICS-D-20-00677R2

**Press Queries**

---

## [Editor Report · Acceptance letter]

19 Nov 2020

PGENETICS-D-20-00677R2 

Involvement of glycogen metabolism in circadian control of UV resistance in cyanobacteria 

Dear Dr Iwasaki, 

We are pleased to inform you that your manuscript entitled "Involvement of glycogen metabolism in circadian control of UV resistance in cyanobacteria" has been formally accepted for publication in PLOS Genetics! Your manuscript is now with our production department and you will be notified of the publication date in due course.

With kind regards,

Nicola Davies

PLOS Genetics

On behalf of:
